# The True Identity of the "Second Pollen Morphology" of *Camellia oleifera*—Stomium Cells

**Yang Hu and Chao Gao ***

Institute for Forest Resources and Environment of Guizhou, Key Laboratory of Forest Cultivation in Plateau Mountain of Guizhou Province, College of Forestry, Guizhou University, Guiyang 550025, China; xfy_gzhy@163.com
* Correspondence: gaochao@gzu.edu.cn

**Abstract:** Previous studies on *Camellia oleifera* pollen morphology have indicated dual morphologies, defined as "dimorphism". However, they were limited to morphological studies at the end of final development and did not elucidate the origin, structure, and function of the second pollen morphology (striate pollen). In this study, the differences between the two "pollen" types were compared by paraffin sections, scanning electron microscopy (SEM), fluorescence microscopy, and in vitro germination. The results clearly showed that the second pollen type was formed by stomium cells of the anther, which is responsible for anther dehiscence. The nucleus and vesicles of the stomium cell were specifically distributed during anther development, which may be related to the formation of the septum, pollen dispersal activity, and the increase in stomium cell count; at the microscopic level, the stomium cell mainly consisted of the cell wall, large vesicles, and nucleus. The large vesicles facilitate the rapid dehydration of stomium cells under suitable conditions for anther dehiscence. Furthermore, studies on other species of the genus *Camellia* have suggested that the second type of pollen morphology is pseudopollen, which is capable of partaking in deceptive pollination. The present study refuted this theory and suggested that the pseudopollen are stomium cells, whose structure relates to their function. These results provide the basis for further research on *C. oleifera* pollen physiology toward the improvement of pollination rates with agricultural practices or breeding interventions.

**Keywords:** *Camellia oleifera*; pollen; stomium cells; dimorphism; pseudopollen; adaptation to pollinators

## 1. Introduction

Pollen is a crucial carrier of genetic information in angiosperms and is released from the pollen sac after anther dehiscence [1–3]. Pollen morphology is strictly genetically regulated, and the pollen of different species or varieties has certain morphological recognition features that are widely employed in studies such as taxonomy or palynology [4,5]. The pollen of most angiosperms has only one morphology, and only a few plants have two types of pollen with similar outer wall ornamentation but different morphologies, which is also called dimorphism. Defining pollen dimorphism is crucial for elucidating the evolutionary direction or reproductive development of the species [6,7]. *Camellia oleifera* is an important species of the *Camellia* genus (Theaceae). Its pollen morphology has always been considered to have two forms, either reticulate or striate. Reticulate pollen is considered to be the most widespread type, whereas striate pollen is much rarer by comparison. Researchers have divided opinions on the second pollen type (hereinafter also referred to as "dimorphic pollen") [8–12]. Whether *C. oleifera* pollen exhibits dimorphism is an important scientific question that can only be addressed through a morphological study [10]. Since the two pollen morphologies of *C. oleifera* are different and do not conform to the definition of plant pollen dimorphism, there is a need for a systematic study on the second pollen morphology of *C. oleifera*.

As a unique and important woody edible oil species in southern China, the reproductive biology of *Camellia oleifera* has been widely studied by Chinese scholars [3,8–14]. Three consenses on *C. oleifera* pollen have been formed in related studies, i.e., the pollen morphology of *C. oleifera* being dimorphic; yield-oriented selection and breeding process of *C. oleifera* reducing the pollen viability of *C. oleifera*; and the main planted *C. oleifera* varieties needing to be regulated to enhance pollen viability.

*C. oleifera* pollen research identified striate pollen using the pellet method [15], scanning electron microscopy (SEM) [16], and in vitro germination [17]. Early studies have shown that *C. oleifera* pollen viability is as high as 83% [18,19]. However, later studies on *C. oleifera* asexual lines showed pollen viability ranging from 40 to 60% [1,10,17,20,21]. It thus appears that the yield-oriented selection in the breeding process reduces the pollen viability of the selected *C. oleifera* varieties. Furthermore, external application of nutrients and hormones can increase pollen viability [20–22], which in turn improves the fruit setting rate of *C. oleifera* [23]. Nevertheless, it is an oversimplification to classify *C. oleifera* pollen as dimorphic solely based on the morphology of the "pollen" dispersed from the pollen sac during pollination, rather than from the perspective of anther development or the internal structure of the two "pollen" types. Therefore, in-depth palynological studies are urgently needed to elucidate the correct terminology for *C. oleifera* pollen types.

A recent study indicated that the second pollen type of *Camellia oleifera* had no germination ability in a solid medium, and the germination rate of *C. oleifera* pollen could reach more than 90% after excluding the second pollen type from total pollen quantity [24,25]. These findings suggest that the inclusion of the second pollen type without germination ability in the total pollen of *C. oleifera* may lead to the conclusion that the germination rate of *C. oleifera* pollen is low. Pollen morphology varies greatly among species, and while similar in external morphology within the same species, it differs in microscopic morphology [5,16,26,27]. Regardless, the second pollen type of *C. oleifera* does not conform to this rule. Since the pollen viability of *C. oleifera* varies widely among different studies, the lack of in-depth pollen morphogenesis research in *C. oleifera* may lead to misjudgment of the pollen's dimorphic nature. This in turn has a direct impact on pollen viability results of previous and future studies and can hamper efforts of pollination optimization. Therefore, this study addressed the following questions: (1) What are the differences between normal pollen and the second pollen type of *C. oleifera*? (2) What are the factors involved in pollen embryogenesis, development, structure, and function of the second pollen type of *C. oleifera*? (3) What are the impacts of the misclassification of the second pollen type on research related to *C. oleifera*?

## 2. Material and Methods

### 2.1. Material Acquisition

Samples were collected at the *C. oleifera* research workstation of Guizhou University during the flowering phase. The sampled plants were managed with regular water and fertilizer. We selected adult *C. oleifera* trees that were healthy, free from diseases and pests, and able to flower and bear fruit normally. The *C. oleifera* varieties evaluated were Changlin 53, Xianglin XLC15, and Huashuo.

### 2.2. Morphology of Anther Dehiscence

The anthers of the Changlin 53 were extracted with forceps, cut crosswise and longitudinally with a double-sided blade, and observed and photographed with a stereomicroscope Leica S9i (Wetzlar, Germany).

### 2.3. Internal Structure of Anther Dehiscence

The collected anthers of *C. oleifera* (Xianglin XLC15 and Huashuo) were extracted and fixed in Carnoy's fluid (ice acetic acid: 95% ethanol = 1:3 (*v*/*v*)), transferred to 70% ethanol solution, and stored in a refrigerator at 4 °C. Hematoxylin staining was performed for sample preparation [28]. After embedding, the samples were sectioned using a slicer (Leica

RM 2235) at a thickness of 8–10 μm. Subsequently, the slices were sealed with neutral resin and photographed with an optical microscope (Leica DM 3000). For convenience, the division of the developmental stages of anthers in *C. oleifera* was performed according to Hu's method (Table 1) [1].

**Table 1.** Classification of the developmental stages of *C. oleifera* anthers.

| Phase | Main Characteristics | Phase | Main Characteristics |
|---|---|---|---|
| 1 | The stamen primordium has only three layers of cells (i.e., inner, medium, and outer) which develop into the connective tissue, sporogenous cells, and epidermis, respectively | 8 | Pollen mother cell completes meiosis I |
| 2 | Sporogenous cell formation, with one layer of anther wall (epidermal cells) | 9 | Pollen mother cells complete meiosis II to form tetrads |
| 3 | Sporogenous cells divide to produce primordial cytoplasmic cells and primordial wall cells with two layers of anther walls | 10 | The callus enclosing the tetrad descends to liberate a single microspore, and the nucleus of the microspore remains in the center of the cell, often called the mononuclear phase |
| 4 | Primary cytoplasmic cells divide to form secondary cytoplasmic cells, and primary wall cells divide to form two layers of primary walls | 11 | The microspores form large vesicles while the nucleus moves to the edge of the cell, often called the mononuclear leaning phase |
| 5 | The development of the anther wall is basic, producing a total of five layers of anther wall, and secondary spore-forming cells begin to divide and proliferate | 12 | The nucleus of the microspore divides asymmetrically, producing a large and a small nucleus for nutrition and reproduction |
| 6 | Production of pollen mother cells surrounded by a common callus | 13 | Microspore development is mature, inner wall cells are radially thickened, and stomium cells are formed |
| 7 | Degradation of callus releases individual pollen mother cells | 14 | Anthers dehiscent from the stomium cells for dispersal of pollen |

### 2.4. SEM Observation of Anther Dehiscence

Individual anthers of *C. oleifera* (Changlin 53 and Huashuo) were cut crosswise and longitudinally using a double-sided blade. To avoid reagents affecting the morphology of the samples, the specimens were observed without fixation on the day of collection and photographed directly under a desktop SEM (TM 4000 plus; Hitachi, Tokyo, Japan).

### 2.5. Fluorescence Observation of Pollen and Second Pollen

Sporopollenin is the main component of the outer wall of pollen, and pectin is the main component of the cell wall. Inorganic acids cannot dissolve sporopollenin but can partially dissolve pectin [29,30]. To distinguish the characteristics of the outer wall of pollen and the cell wall of the second pollen, acid digestion of the anthers of the *C. oleifera* (Huashuo) was performed using dilute hydrochloric acid.

The anthers fixed with Carnoy's fluid were rehydrated using a 70–50–30–0% alcohol concentration and acid digested with 1 M dilute hydrochloric acid in a water bath at 60 °C for 10 min. The acid-digested material was rinsed three times for 10 min using 0.1 M phosphate-buffered saline (PBS), followed by 2 μg/mL 4′,6-diamidino-2-phenyl configured with the same PBS indole (DAPI) fluorescently stained under dark conditions for 10 min. Finally, observation and photography were performed using an optical microscope (Leica DM 3000).

### 2.6. In Vitro Germination of Pollen and Second Pollen

In vitro germination is a common method in the study of *C. oleifera* pollen, which provides a reference for related studies to reject the second pollen type. In vitro germination of normal *C. oleifera* pollen and the second pollen type was performed with reference to the

basic medium reported by He et al. [10]. The germination ability of normal pollen and the second pollen type was checked after 2 h incubation by taking a photograph immediately after spreading Changlin 53 pollen onto the medium using an optical microscope (Leica DM 2500).

### 2.7. Area Characteristics of the Cross-Section of the Stomium Cells

The innermost anthers in *C. oleifera* have four full pollen sacs, and compression was applied to deform the outer layers of the anthers. For paraffin sectioning experiments, the innermost anthers of the 13th phase were selected. The area of stomium cell clusters, individual stomium cells, and individual pollen sacs (without anther walls) was measured using ImageJ v1.8. As only degraded residues of the tapetum remained in the anthers of *C. oleifera* at the 13th phase, the measured area of individual pollen sacs included the area of the tapetum. When measuring the area of individual stomium cells, the cells were selected based on having the largest area (against the anther connective) and a visible nucleus (against the septum); this was to avoid the issue of selecting exceedingly small areas arising from sectioning. At least three anthers were used for all measurements, and three replicates per measurement were averaged. Finally, the data were analyzed using Excel 2010.

## 3. Results and Analysis

### 3.1. Morphology Changes in the Anther Dehiscence Process

Figure 1 shows macro to microscopic images of the different stages of the anther dehiscence process. Early in the flowering phase of *C. oleifera*, only the sepals were visible throughout the flower bud (Figure 1A1). After peeling off the sepals, the anthers inside the bud were close together, with the stigma exposed in the middle (Figure 1A2). The pollen sac of a single anther was yellow in color (Figure 1A3), and the two pollen sacs located on the same side were filled with pollen. The pollen sac junction (also called the stomium) was not yet open (Figure 1A4) but stomium cells had already formed. The stomium cells were more prominent and larger than the adjacent cells under the microscope (Figure 1A5). The flower buds then began to expand to expose the petals. At this point, the anthers began to become exposed (Figure 1B1), the stomia of the anther cluster opened, and the pollen became visible (Figure 1B2). Individual anthers began to lose water and became beige in color (Figure 1B3). The septum was at that point visible at the stomium (Figure 1B4). The cross-sectional images revealed distinct white and nearly transparent cells at the location of the stomium cell cytogenesis, which may be the second pollen type of *C. oleifera*, distinct from the yellow pollen (Figure 1B5). As the petals in the bud of the *C. oleifera* flower further opened (Figure 1C1), the stamen cluster appeared yellow due to anther dehiscence (Figure 1C2) and individual anthers showed obvious dehiscence activity (Figure 1C3). The expanding stomium between the pollen sacs presented white pollen, which is mainly distributed at the two stomia and at the anther wall that is dehiscing (Figure 1C4). Due to the anther dehiscence, only part of the yellow and white pollen can be seen in the cross-sectional image (Figure 1C5). Subsequently, the flower opened completely (Figure 1D1) and the surface of the stamen cluster was covered with pollen grains (Figure 1D2). The individual anther was completely dehiscent and white pollen could be seen at the edge of the stomium in the anther wall (Figure 1D3). A large amount of white pollen was distributed at the stomium (Figure 1D4), and the dehiscence angle of the pollen sac was greater than 180 degrees (Figure 1D3,D5). We found white and yellow pollen grains during the dehiscence process of the *C. oleifera* anther; the white grains were considered the second pollen type in related studies. However, the white pollen was found at the location of stomium cell cytogenesis, which may be related to stomium cells.

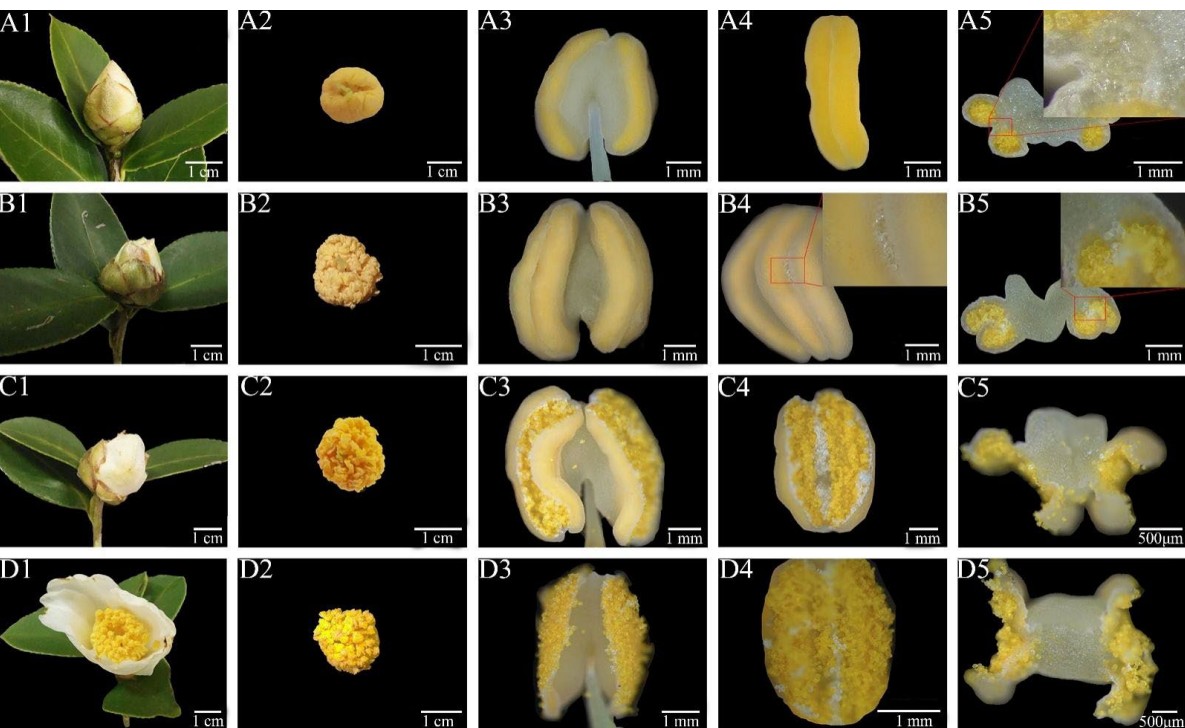

**Figure 1.** Single flower opening and anther dehiscence in *Camellia oleifera.* (**A1–D5**): cross-sections of flower buds, stamens, single anthers, anther stomia, and anther cross-sections of *C. oleifera* in the early flowering phase (**A1–A5**), when sepals but not petals unfolded (**B1–B5**), when petals were exposed (**C1–C5**), and when blooming (**D1–D5**).

### 3.2. SEM Observations of Anther Dehiscence Process

To further explore the relationship between white pollen grains and stomium cells, as well as the difference between white pollen and yellow pollen grains, SEM analysis was conducted. When the anther was at the 13th stage (Table 1), the endothecium was radially thickened. The pollen sac was not yet cracked, but the stomium cells started to differentiate and develop at the location of anther stomium region. In addition, the wall of the stomium cells appeared to be striate. The red circle in the Figure 2 is the stomium cells clusters, with large morphological differences from neighboring cells (Figure 2A1,A2). The septum could be seen on the periphery of the stomium cells, and the outer stomium cells were close to the septum. As dehiscence was reached, their structure showed obvious shrinkage and signs of water loss (Figure 2A3,A4). In the process of bud opening and petal opening, the anthers began to make direct contact with the surrounding atmosphere (Figure 1B1), leading to the dehiscence of the two pollen sacs on the same side of the stomium (Figure 2B3), and the scattering of pollen grains and stomium cells. The size difference between individual stomium cells was obvious, which resulted from outward-leaning stomium cells being subjected to dehydration and becoming smaller in contact with the external environment (Figure 2B4). The stomium cells were mainly distributed at the location of cytogenesis of stomium cells and on the anther wall that split from the above location. Based on the location and morphology of the second pollen type, we believe that the white pollen grains were stomium cells (Figure 2B1,B2,C1). The stomium cells in fully open pollen sacs followed the same distribution pattern, i.e., the stomium cells were distributed toward the surface of the pollen grain population (Figure 2C2,C3). Thus, the stomium cells were scattered on the carrier table (Figure 2C4), and fresh pollen mostly adhered to the dehydrated stomium cells (Figure 2D1). However, they differed significantly in the outer wall ornamentation. *C. oleifera* pollen had at least one germinatin furrow—it always has three, but on the illustrations at least one was visible—and reticulate outer wall (Figure 2D2), whereas stomium cells did not have germination grooves and had striated walls (Figure 2D3). The stomium cells undergoing severe dehydration were also smaller (Figure 2D4). Moreover,

the stomium cells had the same morphology as the second pollen type of *C. oleifera* previously reported, which indicated that the second pollen type originated from the stomium cells.

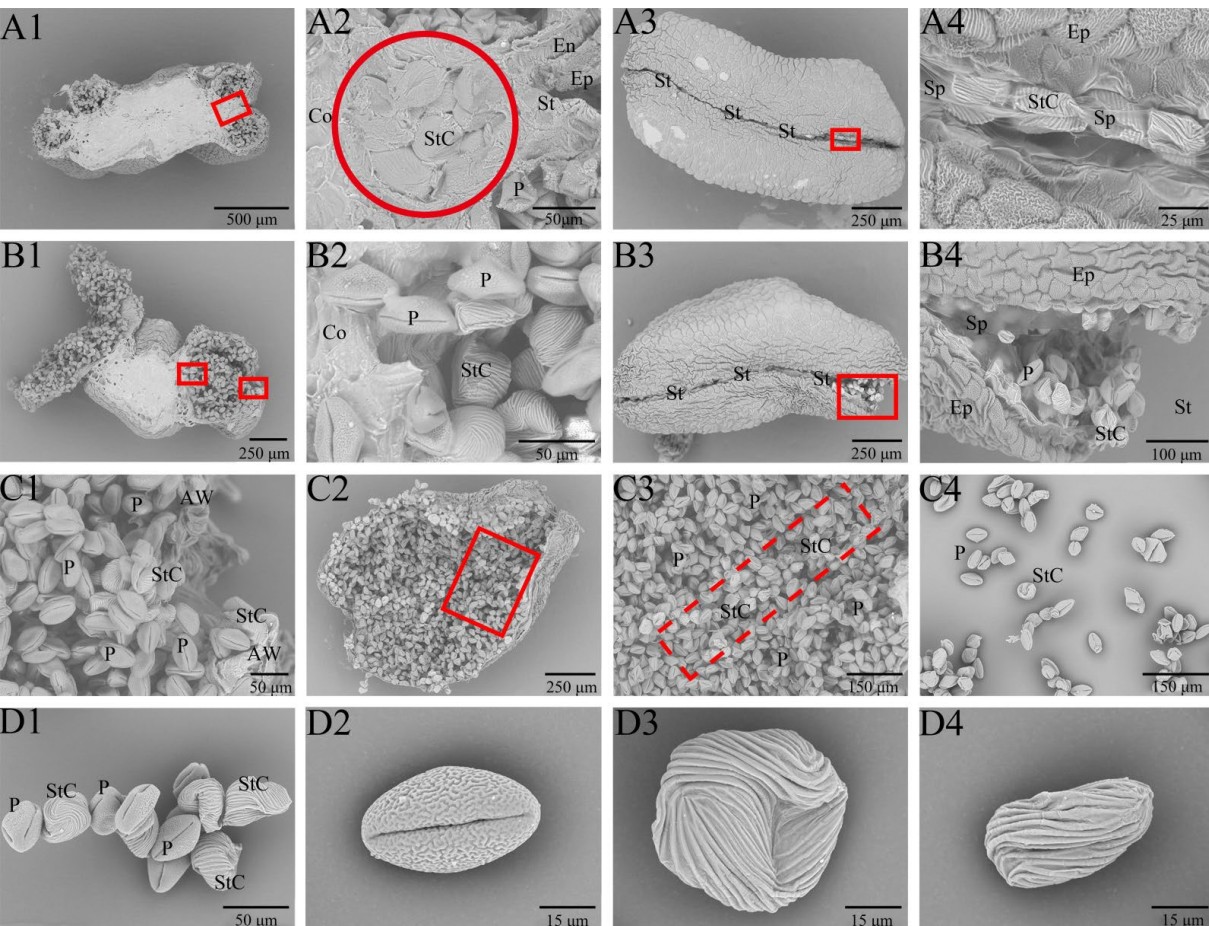

**Figure 2.** SEM observation of anther dehiscence in *Camellia oleifera*. (**A1**): cross-section of *C. oleifera* anthers when they were not dehiscent; (**A2**): enlargement of the red box in (**A1**), and the red circle was a cluster of stomium cells, its cell wall with a striate shape. During this phase, the endothecium was not radially thickened in the stomium cells region, and the pollen grains were sunken inward; (**A3**): the stomium of the anther about to be dehiscent; (**A4**): enlargement of the red box in (**A3**), where the septum was outside of the stomium cells. The stomium cells and epidermal cells were then subjected to dehydration to pull the septum. (**B1**): anther dehiscence; (**B2**): enlargement of the left red box in (**B1**), where a large number of stomium cells were located at the site of stomium cell cytogenesis; (**B3**): anther dehiscence due to septum dehiscence at the stomium, where stomium cells and epidermal cells were subjected to dehydration for dehiscence; (**B4**): enlargement of the red box in (**B3**), with the anther dehiscing at the stomium. The stomium cells and pollen were mixed at the stomium. (**C1**): enlargement of the right red box in (**B1**), where some of the labial cells stuck to the anther wall and adhered to the pollen against the outside with dehiscence activity; (**C2**): anther dehiscence, with the stomium exposing a large number of pollen grains and stomium cells inside; (**C3**): enlargement of the red box in (**C2**), where the stomium cells were distributed under the stomium at the connection of two pollen sacs (inside the red dashed box), which was also the location of stomium cell cytogenesis; moreover, stomium cells were mostly located on the surface of pollen; (**C4**): pollen and stomium cells mixed after sprinkling pollen on a conductive gel. (**D1**): adherence of pollen and stomium cells; (**D2**): the equatorial surface of pollen, where the outer wall of pollen was reticulate, exposing a germination furrow; (**D3**): unhydrated stomium cells with the striate cell wall and a larger volume than pollen grains; (**D4**): hydrated stomium cells with a smaller volume than pollen grains. AW, anther wall; Co, connective; En, endothecium; Ep, epidermis; P, pollen; Sp, septum; St, stomium; and StC, stomium cells.

### 3.3. Microscopic Observation and Fluorescence Observation of Stomium Cell Cytogenesis during Anther Dehiscence

At the 10th phase of anther development (uninucleate-medium phase) (Figure 3A1), deeply stained cells appeared at the base of the stomium, and the adjacent cells within the anther were actively dividing, indicating that the endothecium cells within the anther had dividing activity at the stomium (Figure 3A2). At the 11th phase of anther development (uninucleate-leaning stage), stomium cells appeared at the stomium position—all possessing large vesicles and being larger than the adjacent cells (Figure 3A3). Moreover, the nuclei of the stomium cells toward the exterior of the stomium were distributed extremely regularly—all located toward the direction of the stomium. The nuclei of the inner stomium cells were located in the direction of the connective, and the stomium cells were separated from the pollen inside the pollen sac by a thin middle layer and the tapetum (Figure 3A4).

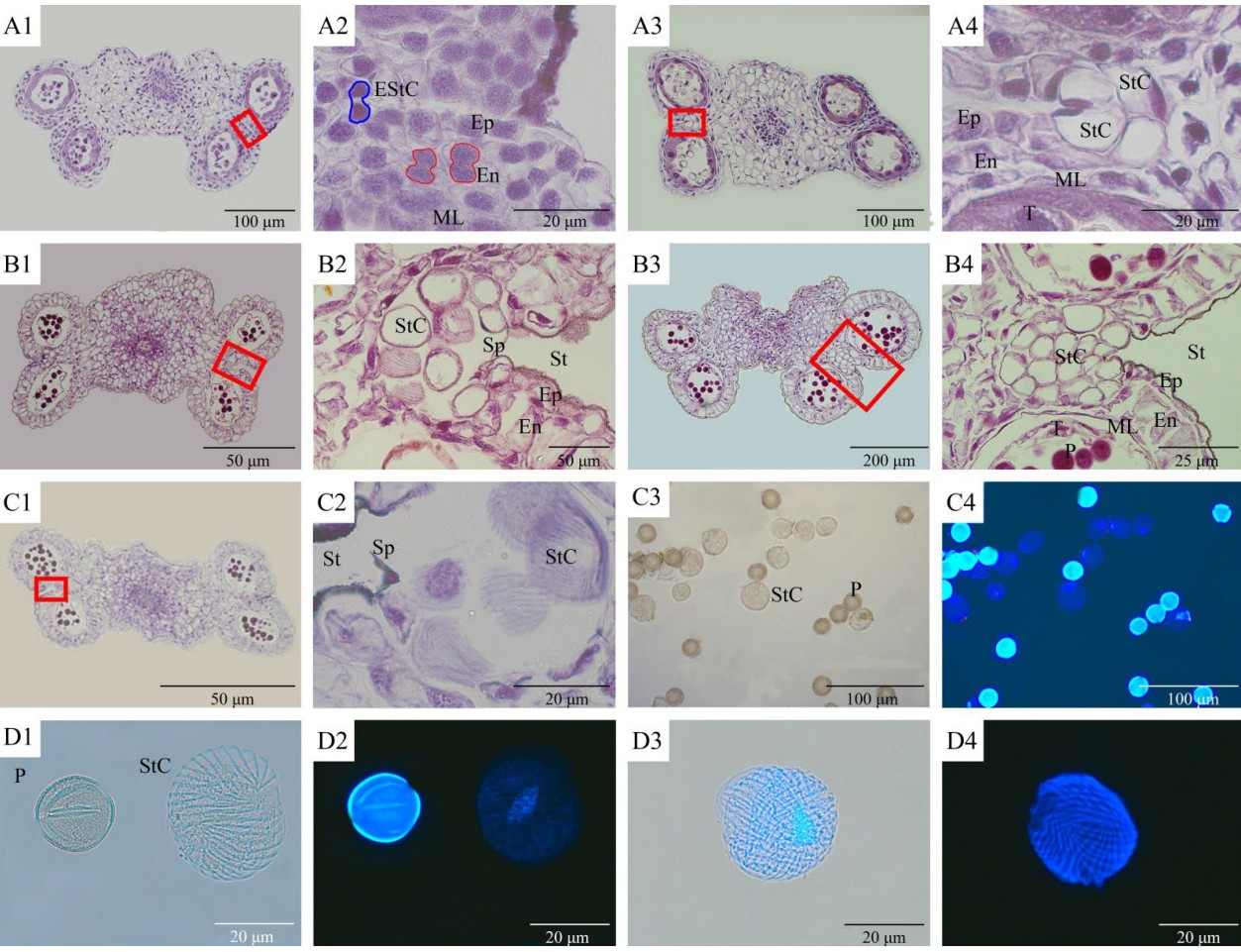

**Figure 3.** Development and structure of stomium cells in anther of *Camellia oleifera*. (**A1**): cross-section of anther at the 10th stage (uninucleate); (**A2**): enlargement of red box in (**A1**), where the nuclei of early stomium cells were stained darker than surrounding cells (marked blue) and endothecium cells were in a divided state (marked red); (**A3**): anther uninucleate-leaning stage, where the stomium cells developed further with large vesicles; (**A4**): enlargement of red box in (**A3**), where the nuclei of stomium cells at the stomium faced the stomium, while the nuclei of stomium cells in the direction of the anther connective faced toward the anther connective, with fewer stomium cells. (**B1**): the stomium cells in the anther were mature, and the endothecium cells did not undergo radial thickening at the stomium; (**B2**): enlargement of the red box in (**B1**), where the cell wall of mature stomium cells had a striated structure, the nucleus faced the connective, the septum was formed, and the cell wall

of the epidermal cells was crumpled; (**B3**): there was a large difference in the stomium cell count between the stomia of the same anther; (**B4**): the stomium cells matured as the pollen developed, and stomium cells on the outside were smaller in size than those on the inside due to faster water loss, making the volume smaller than that of the stomium cells against the interior. (**C1**): late anther development; (**C2**): enlargement of the red box in (**C1**), where the septum of the stomium opened, and dehydration of the stomium cells against the stomium made the volume of the vesicles smaller, resulting in a visible nucleus and less dehydration of the stomium cells against the connective; (**C3**): the stomium cells were lighter in color than the pollen grains under the light microscope, exposing a striate shape with a larger volume change; (**C4**): fluorogram of (**C3**) by DAPI staining, where pollen grains showed bright fluorescence, whereas stomium cells showed the approximate outline and exposed the stained nucleus. (**D1**): an obvious germination furrow in the pollen grains, as well as the striate cell walls and protoplasts after hydrochloric acid digestion of stomium cells; (**D2**): fluorogram of (**D1**) by DAPI staining, where pollen walls emitted bright fluorescence, whereas the stomium cells showed the outline of the protoplasm and the nucleus; (**D3**): fluorescence excitation diagram under bright field after staining of the stomium cells, where the position of the nucleus and the striate cell wall could be seen; (**D4**): acid digestion led to the destruction of the protoplasm of the stomium cell, resulting in the stomium cell being left with only a striate cell wall with visible texture. En, endothecium; Ep, epidermis; EStC, early stomium cells; ML, middle layer; P, pollen; Sp, septum; St, stomium; StC, stomium cells; and T, tapetum.

At the 13th phase of anther development (binucleate stage), the wall of the endothecium thickened radially, but not at the stomium region, which facilitates anther dehiscence (Figure 3B1). A septum appeared on the outside of the stomium cells, and the nuclei of all stomium cells were uniformly distributed in the direction of the anther connective, which made it easier for the intracellular vesicles to lose water by having a larger contact area with the environment. The innermost stomium cells had a cross-sectional area of $320.54 \pm 5.76$ μm$^2$ with a coefficient of variation of 0.02. The extremely small coefficient of variation indicated that the size of the stomium cells was strictly regulated by genetic factors. As for the stomium cells with water loss, the cross-sectional area was reduced by ~2/3 to $121.37 \pm 15.70$ μm$^2$, with a coefficient of variation of 0.12. This large coefficient of variation indicated that the efficiency of water loss in stomium cells varied depending on anther-influencing factors such as environmental humidity exposure (Figure 4). The microstructure of the walls of stomium cells appeared striate, and the stomium cells in the interior were close to the unthickened anther walls (Figure 3B2). The unthickened anther wall also exhibited a striate appearance after the stomium cells detached (Figure 2B2), suggesting the formation of a chimeric pattern by the anther walls and stomium cell walls. The morphological and structural changes occurring due to water loss in the stomium cells (Figure 2D3,D4 and Figure 3B2,C2) may apply a force on the unthickened anther wall, causing the anther to split at the stomium.

The number of cells at the two stomia of the same anther varied considerably, with 8 cells on the left side of the cross-sectioned anther and 18 on the other (Figure 3B3). The differences in the cross-sectional area between stomium cells in different positions before dehydration were not obvious nor statistically analyzed, as the image showed only one section, creating the illusion that differences exist (Figure 3B4). As the anthers further developed, the stomia became smaller in size and the septum was the first to split, with the stomium cells against the exterior, more dehydrated and presenting with visible nuclei. In contrast, the stomium cells against the interior were less dehydrated and able to retain their morphology (Figure 3C1,C2). Pollen grains of *C. oleifera* at the dispersal phase were bright under a fluorescence microscope (Figure 3C3,C4), exposing obvious germination furrows (Figure 3A1,D2). The size difference between pollen grains was not obvious. The stomium cells were dark under dark-field fluorescence, with unclear cell wall ornamentation (Figure 3D2). Size differences were evident, with nuclei faintly visible (Figure 3C3,C4). However, under bright-field fluorescence imaging, the nucleus and the cell wall were both present (Figure 3D3). The cell wall of some of the stomium cells was acidolyzed, allowing the internal protoplasm to

flow out and leaving the unacidified components of the cell wall. At this point, a clear cell wall ornamentation could be observed (Figure 3D4).

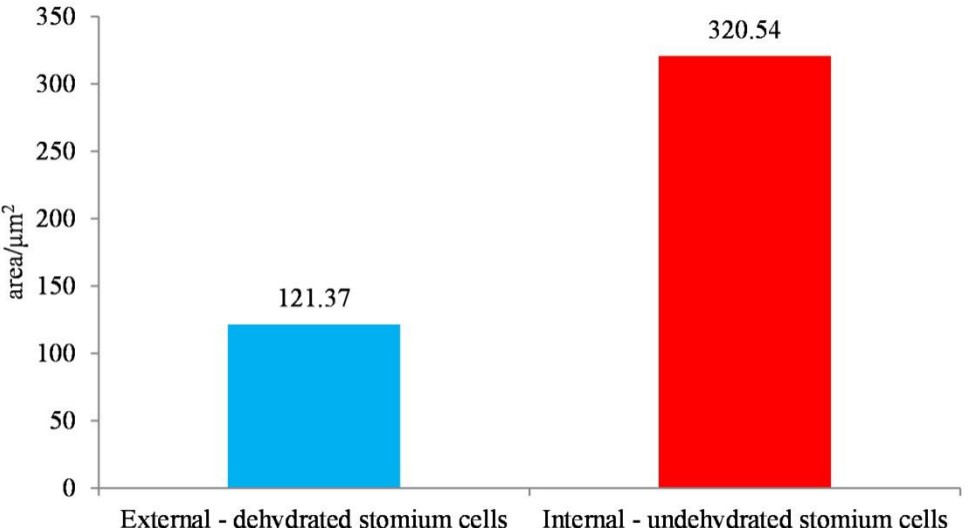

**Figure 4.** Change in cross-sectional area of stomium cells of *Camellia oleifera* during water loss.

*3.4. In Vitro Germination Ability of Pollen and Stomium Cells*

When pollen was spread evenly on the medium, the stomium cells easily stuck together with the pollen grains (Figure 5A). However, stomium cells cannot germinate. Therefore, if the stomium cells are not excluded when counting either the germination rate or pollen number, a relatively low germination rate is obtained. Pollen formed through normal development was able to grow pollen tubes, but some pollen did not germinate (Figure 5B). The reason for non-germination could be the non-viability or the short germination time.

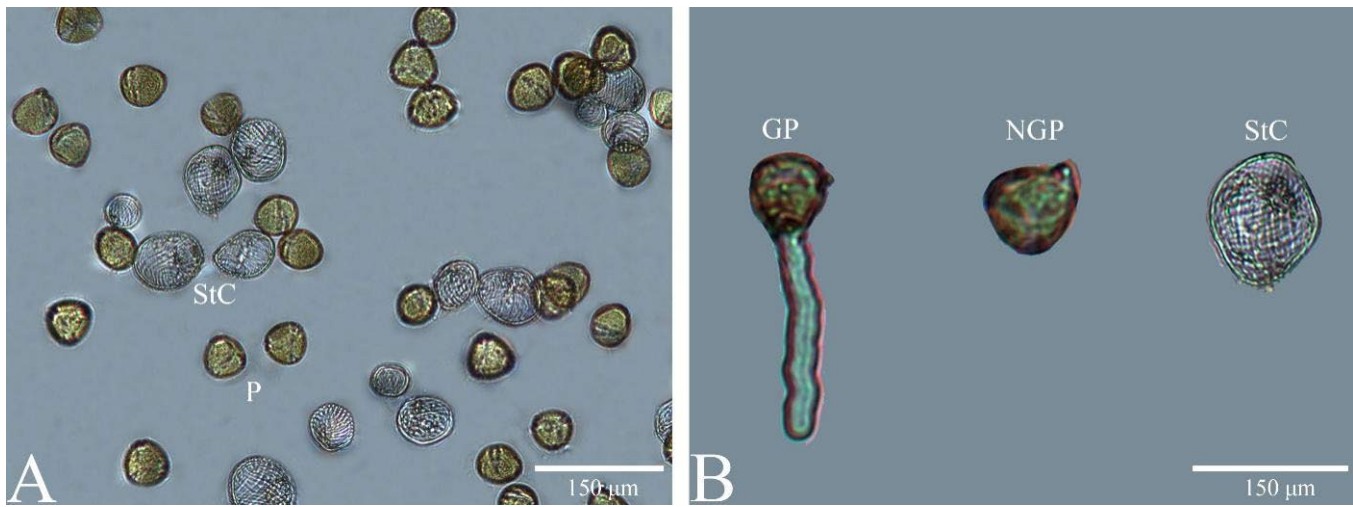

**Figure 5.** In vitro germination of pollen and stomium cells in *Camellia oleifera*. (**A**): a large number of stomium cells mixed with pollen when the medium was first sprinkled with pollen. (**B**): two hours after sprouting, pollen partially germinated, while no sign of germination was found in stomium cells. P, pollen; GP, germinated pollen; NGP, non-germinated pollen; and StC, stomium cells.

## 4. Discussion

As plant pollen morphology is strictly controlled by genetic factors, different species or varieties exhibit distinct morphologies. Therefore, plant pollen morphology is often used as a basis for analyzing phylogenetic relatedness [4,27,31]. Under natural conditions, the morphology of the vast majority of plant pollen is stable over a certain period. How-

ever, in a small number of plant species, pollen morphology exhibits dimorphism. This phenomenon has been explained as a transitional stage of plants in different evolutionary directions [6,7]. Pollen dimorphism occurs in the pollen of complete and incomplete flowers in *Thalictrum macrostylum* [6], and a small amount of pollen in male and female flowers of *Vitis vinifera* [32,33]. On the other hand, as a perfect flower with complete floral organs, the reports of dimorphism in *C. oleifera* pollen are different from the dimorphism produced by non-functional pollen. Pollen dimorphism has also been reported in pollen of *Aconitum gymnandrum* which adapts to pollination conditions and *Campanula americana* pollen with heat stress adaptation [6,33]. However, pollen with clear dimorphism is less reported or lacks clear illustration. For instance, one study suggested that the pollen morphology of *Arnebia szechenyi* is dimorphic without providing reliable pictures [34]. In addition, the pollen of *Ephedra trifurca* was considered to be dimorphic in an earlier study [35]. However, a more recent study showed its dimorphism to be the result of pollen deformation caused by fixation and dehydration steps during sample preparation [36].

Dimorphism of *C. oleifera* pollen was first reported by Chinese scholars in 1986, who discovered that *C. oleifera* has both striate and normal pollen by using the pellet method [15], which contributed to the reproductive biology of *C. oleifera*. However, no further reports on the origin and function of dimorphism in *C. oleifera* pollen have been made since then, leaving the unopposed view that *C. oleifera* pollen possesses dimorphism [16,27,37]. The present study aimed to fill this gap from the perspective of the origin, development, morphology, structure, and function of the second pollen type by demonstrating that the striate pollen is a stomium cell responsible for anther dehiscence and mixed with pollen grains during pollen shedding. Thus, this study demonstrated that *C. oleifera* pollen is not dimorphic and that previous reports misclassified stomium cells as "the second pollen" type. Pollen of the same species in different regions or varieties still has unique characteristics. However, these characteristics have only subtle morphological differences, which are the result of geographical or selective breeding processes [4,38]. The findings of this study are crucial for research related to *C. oleifera*, especially for in vitro germination of *C. oleifera* pollen. The inclusion of stomium cells that cannot germinate into the total number of pollen directly affects the statistical results of the germination rate.

In previous reports of in vitro germination of *C. oleifera* pollen, the statistical methods for stomium cells and pollen grains of *C. oleifera* were hardly illustrated [10,21,39,40]. Some studies had discrepancies between the labels used in illustrations of anther stomium cells in *Camellia* spp. (Table 2). Tsou (1997) defined the second pollen of three *Camellia* spp. as "deceptive to pollinators" and called it pseudopollen [41]. Pseudopollen has been widely reported in orchids, especially epidendroid orchid that does not secrete nectar. These species have evolved pseudopollen with nutrients to provide food rewards to pollinators [42]. While the subject of Tsou's study, *Camellia sinensis*, may be a nectariferous species [43], in his discussion he stated, "The bowl-like flowers of Camellioideae have numerous showy stamens, and pollen grains serve as the sole or primary pollination reward . . . " the histochemical test for starch, protein, and lipids on pseudopollen was negative. This indicates that pseudopollen contains no or extremely few nutrients [44]. However, pseudopollen from orchids contains protein and a small amount of starch [45]. In addition, the *Camellia* "pseudopollen" theory cannot explain the structural changes we found in "pseudopollen". This proves that the definition of stomium cells as pseudopollen is inaccurate. Due to the spatial location of stomium cells, it would inevitably stick to pollinating insects collecting nectar. However, the pollinating insects do not purposely collect stomium cells. Therefore, it cannot be ruled out that the stomium cells may have a certain function of diluting the pollen to improve the pollination success rate with the actual pollen.

For *C. oleifera*, the nectar secretion of single flowers was 145.40 ± 24.89 μL during 24 h of flowering, and 421.20 ± 14.00 μL throughout the flowering period, with an average sugar content of 29.12 ± 0.94%, and containing 17 amino acids. When *C. oleifera* flowers were pollinated under natural conditions with the stamens removed (only nectar produced and no pollen), the fruiting rate was not significantly different from that under natural pol-

lination conditions (both pollen and nectar), indicating that nectar is the pollination reward to pollinating insects [12]. Therefore, *C. oleifera* does not need pseudopollen to provide additional food for pollinators. Furthermore, defining cells in this region as pseudopollen leads to the conclusion that the stomium cells responsible for anther dehiscence are not present in the *C. oleifera* anthers. This notion that pseudopollen is occurring in *Camellia* spp. has caused confusion and disparity in the findings of flower botanical and development studies in the genus [46].

**Table 2.** Part of the misjudgment or explanation about the research on the stomium cells of *Camellia* anthers.

| Species | Labeling or Description | Literature |
|---|---|---|
| *C. gauchowensis* | Types | [16] |
| *C. magniflora* | Abnormal pollen | [47] |
| *C. sinensis* | Pseudopollen | [41] |
| *C. tenuifolia* | Pseudopollen | [41] |
| *C. oleifera* | Empty pollen | [48] |
| *C. oleifera* | false pollen | [17] |
| *C. oleifera* | Male sterile pollen (with photo) | [37] |
| *C. oleifera* | Striate pollen | [15] |
| *C. oleifera* | Pseudopollen | [24] |

During anther dehiscence, stomium cells are mostly distributed within the surface layer of pollen. Therefore, the mixing of pollen would occur readily in various experimental operations. Deng et al. (2020) included stomium cells in the total number of pollen when counting pollen viability, but additionally counted the proportion of stomium cells, resulting in maximum pollen viability of only 56.23%; the proportion of stomium cells (pseudopollen) and abnormal pollen reached ~15% [17]. On the other hand, in a different study, when the stomium cell count was excluded from the total calculation, pollen viability reached a maximum of 93.02% [24,25]. In the $F_1$ cross between Youxian *C. oleifera* and Huashuo, the highest ratio of pollen grains to stomium cells produced by a single plant was 1.7 [24]. In this case, if the stomium cells were included in the total number of pollen, the germination rate would be only 62.96% even if all the real pollen germinated, thus the true germination rate would be miscalculated. These results suggest that some of the previous studies on *C. oleifera* pollen viability underestimated the viability of *C. oleifera* pollen. Nevertheless, studies on the gain of pollen viability by applying nutrient elements [21], vitamin C [20], and hormones [22] are still reliable.

Different species have different methods of stomium cell development. In *Arabidopsis thaliana*, the stomium region is differentiated during anther development. The stomium region is mainly composed of a septum tissue that separates two anther sacs and a stomium cell responsible for anther dehiscence. The anther becomes bilocular after degeneration and breakage of septum below the stomium as it matures [2]. The septum dehisces but the stomium cells do not, creating two pollen sacs that are connected but not in contact with the external environment. Subsequently, the stomium cell dehisces to release pollen [49]. In *Nicotiana tabacum*, during the early stages of anther development, the epidermal cells during the dehiscent region are structurally distinct from the surrounding cells as evidenced by SEM and they develop directionally into stomium cells and round cell clusters, respectively. The round cell cluster dehisces before the stomium cell, allowing the two pollen sacs to be connected [50]. Stomium cells of *C. oleifera* are difficult to distinguish from neighboring cells at the microscopic level during ontogenesis. Whether the stomium cells develop from ontogenetically specific cells requires further observation at the SEM level.

When the anthers of *C. oleifera* reached the 10th phase, a group of darker stained cells arose at the base of the stomium. The adjacent cells were in a state of division and a large vesicle was formed in these cells by the 11th phase. The presence of large vesicles is one of the characteristics of a mature labellum cell. When the plant cell divides, the vesicles often split into many small vesicles. As the small vesicles are not obvious under optical microscopy, cells with large vesicles are often not actively dividing [51]. However, the

stomium cell count increased significantly at the 13th phase [1]; therefore, this may not be the result of cell division. During the 11th phase, the nuclei of the stomium cells next to the anther connective also faced the anther connective. Further investigation is required to verify whether the nuclei of the stomium cells transmit information to cells in the adjacent anther connective region, inducing adjacent cells to the stomium cell transformation.

Furthermore, during the 13th phase, the nuclei of the stomium cells near the stomium faced towards the stomium and form a septum—suggesting that the external stomium cells play a role in the formation of the septum. The septum ruptured after pollen maturation, and before that, the nuclei of the stomium cells faced uniformly toward the connective, allowing a greater surface area of the large vesicles towards the stomium and making it easier for the large vesicles to dehydrate. Dehydration shrunk the stomium cells to create traction on the surrounding cells and allowed the anther walls to split at the stomium. This mechanism facilitates pollination on sunny days when pollinators are active, making it easier for *C. oleifera* to pollinate and set fruit [52], while in contrast, during cloudy and rainy days, anther dehiscence remains incomplete [37]. Therefore, this kind of dehiscence is an adaptation of the stomium cells during the long-term evolutionary process, which allows the *C. oleifera* anthers to rapidly dehydrate under suitable conditions to open the pollen sacs until complete pollen dispersal. Importantly, Tsou's "pseudopollen theory" cannot explain the structural changes of this cell type. Therefore, our findings on structural changes during anther dehiscence justify our proposal that these cells are stomium cells.

The stomium cells of *C. oleifera* that are in direct contact with the external environment for rapid dehydration and deformation cause the unthickened region of the endothecium to undergo dehiscence. In contrast, mutant anthers of *A. thaliana* fail to undergo dehiscence as they lack a secondary thickening of the endothecium [53]. Anther dehydration and thickening within the endothecium are important for dehiscence [54]. However, descriptions of anther dehydration focus mainly on the 'epidermal layer dehydration' and not the stomium cell dehydration. An important reason for this is the small area comprised of stomium cells in plants such as *A. thaliana*, *N. tabacum,* and *Oryza sativa*; in the anther structure model, the area of the stomium cell in *N. tabacum* was ~2676.90 $\mu m^2$ [2,50,55]. Therefore, the overtly small area of the stomium cell in these plants often makes their role negligible, while the stomium cells of *C. oleifera* can take up to 14,410.75 $\mu m^2$ of area, representing 30.92% of the total area of an individual pollen sac. Existing biomechanical studies of anther dehiscence have focused on the role of water loss from the anther epidermis and thickening within the endothecium [54]. The epidermis of mutant anthers of *A. thaliana* remains unaffected by drought treatment and does not exhibit a contracted phenotype; however, drought treatment could still cause anthers to split at the stomium and release pollen [56]. *A. thaliana* anthers with a smaller stomium cell area can still undergo dehiscence without epidermal contraction. The larger area of the stomium cells in *C. oleifera* suggests that existing theories cannot explain the biomechanics of its anther dehiscence, thus further in-depth research is required.

Stomium cells within the stomium cell cluster differ in dehydration based on their exact location. Stomium cells in the interior are morphologically larger than pollen grains due to slower dehydration. Stomium cells after dehydration are morphologically more variable and can appear smaller than pollen grains [17]. Furthermore, pollen walls of different species can emit different fluorescence [57,58]. The main component of the pollen wall is sporopollenin, which is extremely resistant to non-oxidizing chemicals and enzymes [59]. However, the main component of the cell wall is pectin, which can be degraded under weak acid conditions [30]. The present study showed a difference in composition between pollen walls and stomium cell walls. Under acidolytic conditions, the outer pollen wall retained its structure and fluoresces. However, once the stomium cell wall was disrupted, part of the stomium cell protoplasm disappeared, leaving behind the acidolyzed cell wall, resulting in a striated appearance under fluorescence. DAPI is widely used for chromosome staining and exhibits intense fluorescence when it binds specifically to AT sequences in bases [60]. In addition, DAPI can also label microtubules and microfilaments in cells [61]. However,

whether microtubule material is present in the striate cell wall of stomium cells as well as its compositional similarities to the outer wall of pollen still needs further investigation. The cell morphology of stomium cells has not been reported. However, the striate morphology of stomium cells in *C. oleifera* is close to the type of branched helices with thickened inner walls in Solanaceae plants [62], which may provide some direction to investigate the origin of stomium cells in *C. oleifera*.

The morphology of *C. oleifera* pollen has long been considered to be dimorphic. However, this study showed that the "second pollen" type of *C. oleifera* is actually the stomium cells, which anatomically control anther dehiscence and mix with the true pollen during dehiscence. A major differentiator is that the outer wall of *C. oleifera* pollen has reticulate ornamentation instead of striate ornamentation observed in stomium cells. The true pollen can germinate whereas the stomium cells, by definition, cannot. Therefore, stomium cells should no longer be counted as pollen in determining the germination rate and quantity of pollen. This may further prevent the low viability reporting of *C. oleifera* pollen. Nevertheless, studies that reported on the improvement of *C. oleifera* pollen viability by certain elements (i.e., nutrients, hormones, etc.) are still reliable. The nucleus and vesicles of stomium cells appear to have certain developmental patterns related to the dehiscence of *C. oleifera* anthers, showing stomium cell functions specific to anther structure. In conclusion, this study corrected the view of the morphology of *C. oleifera* pollen by comparative analysis of *C. oleifera* pollen and stomium cells which provides a basis for further research of *C. oleifera* pollen physiology toward the improvement of pollination rates with agricultural practices or breeding interventions.

**Author Contributions:** Conceptualization, Y.H.; methodology, Y.H.; software, Y.H.; validation, Y.H.; formal analysis, Y.H.; investigation, Y.H.; resources, C.G.; data curation, Y.H.; writing—original draft, Y.H.; Writing—review & editing, C.G.; visualization, C.G.; supervision, C.G.; project administration, C.G.; funding acquisition, C.G. All authors have read and agreed to the published version of the manuscript.

**Funding:** This work was supported by the Science and Technology Plan Project of Guizhou Province of China (Qian Ke He Zhi Cheng [2022] Zhong dian 017 Hao), National Natural Science Foundation of China (32060331), and Cultivation Project of Guizhou University ([2019]35).

**Data Availability Statement:** The data used for the analysis in this study are within the article.

**Acknowledgments:** We thank Shikui Lu and Quanen Deng for their assistance in sampling.

**Conflicts of Interest:** The authors declare no conflict of interest. The funders had no role in the design of the study; in the collection, analyses, or interpretation of data; in the writing of the manuscript, or in the decision to publish the results.

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
