# Peer review of "The True Identity of the “Second Pollen Morphology” of Camellia oleifera—Stomium Cells"

_horticulturae, doi:10.3390/horticulturae8040347_

Round 1
Reviewer 1 Report
The manuscript is well written, complete, of high quality, with information necessary for the proposed study. I suggest minor modifications before publication (see attached pdf).

Reviewer 2 Report
Ref. The True Identity of the "Second Pollen Morphology" of Camellia oleifera - Stomium Cells
The paper is a kind of descriptive study of the pollen grains morphology and biology (germination) in Camellia oleifera, an important woody edible oil species in southern China.
The ms is easy to follow, it is good designed and carefully structured. Figures (photographs) are of very good quality.
My only one concern relates to the definition of the ‘second pollen" at the beginning of the text – in Introduction section. Please, clarify (in brief) more specifically ‘second pollen’ term. For me the term is a bit confusing.
Reviewer 3 Report
This is a very interesting and technically perfectly done paper. All findings are documented by excellent illustrations arranged in plates that allow a quick and perfect survey on the subject. However, the text suffers from various shortcomings. It is sometimes difficult to follow because of two reasons. First the English needs a careful revision by a native speaker. Second, the terminology used is not the standard terminology used in plant morphology and palynology. Only the trained botanists can extract from the text with the help of the excellent illustrations what the authors really mean. There are also some minor misunderstandings of the bibliography, most likely also based on wrong understanding of the terminology applied there.
The only thing that does not become really clear is the origin of the pollen like structures that could be clearly termed pseudo pollen. They originate from a furrow of the anther wall along the dehiscence line. They could be seen as epidermal hairs and it would be interesting if any epidermal cell forms only a single one of these cells or if it is a (short) moniliform chain of such cells. In some species pseudo pollen is the only structure to attract pollinators to avoid that they feed at all on pollen. In others, the pseudo pollen just “dilutes” the pollen to keep pollinators longer on the flower or to allow successful pollination with a smaller total amount of real pollen. Non functional pollen in dioic species (e.g. in female flowers of Actinida) should not be mixed up with dimorphic pollen.
I made comments in the attached PDF. The place where I made the comments is not necessarily the first or only place where the problem occurs. It is just the place where I got the problem as such and suggested a solution.
In general, the paper clearly needs a thorough revision of the text. As it is otherwise wonderful and technically perfect, it is well worth the effort.

Round 2
Reviewer 3 Report
Point 1: I did not argue against „dimorphism“, but against the duplication „morphological dimorphism“. All your examples are examples for the use of “dimorphism” but in none appears “morphological dimorphism”. So delete “morphological” here.
Point 2: “Dimorphic pollen” is clearly a plural as one pollen grain cannot be dimorphic. You do not mean “dimorphic pollen” here but the second pollen type. A correct phrasing would be: “The results clearly show that the second pollen type is in fact formed by stomium cells of the anther.” This and other things illustrate that your native speaker was not a botanist.
Point 6: your explanation is fine, but I can see that from the pictures. The description “anther dispersal” is wrong anyway as the anthers are not dispersed. You mean obviously pollen dispersal.
Point 7: the “second pollen” in your text is a singular while you definitely do not mean an single grain. Write “the white grains are considered the second pollen type in related studies.
Point 11. The pollen has not “at least one germinatin furrow, it has always three but on the illustrations at least one is visible.
Point 15: I am familiar with microsporogenesis and know that pollen is distributed either in a two cell stage (pollen tube cell) and spermatogenous cell or at a three nucleate stage with pollen tube cell and two sperm cells. I suggest to read standard textbooks on embryology like Johri.
Point 17: “programmed cell death” is generally used if the genetic program or process is known. Here the septum is most likely just ruptured due to the fact that it is not able to follow the growth of the anther. Describe what you see and do not speculate about reasons that are not studied.
In addition, under 3.2. appears the phrasing “the pollen wall of the stomium”. The stomium may have a wall but it is not the pollen wall.
Please go for a native speaking botanist!
Author Response
请参阅附件。
